# The Blood Plasma Lipidomic Profile in Atherosclerosis of the Brachiocephalic Arteries

**DOI:** 10.3390/biomedicines12061279

**Published:** 2024-06-09

**Authors:** Anastasiia Lomonosova, Daria Gognieva, Aleksandr Suvorov, Artemy Silantyev, Alina Abasheva, Yana Vasina, Magomed Abdullaev, Anna Nartova, Nikolay Eroshchenko, Viktoriia Kazakova, Roman Komarov, Andrey Dzyundzya, Elena Danilova, Dmitry Shchekochikhin, Philipp Kopylov

**Affiliations:** 1World-Class Research Center «Digital Biodesign and Personalized Healthcare», I.M. Sechenov First Moscow State Medical University (Sechenov University), 119048 Moscow, Russia; gogniva_d_g@staff.sechenov.ru (D.G.); suvorov_a_yu_1@staff.sechenov.ru (A.S.); artsilan@gmail.com (A.S.); alina.abasheva@mail.ru (A.A.); yv.medstudent@gmail.com (Y.V.); magomed19942@mail.ru (M.A.); nartovaanna@mail.ru (A.N.); kopylov_f_yu@staff.sechenov.ru (P.K.); 2Laboratory of Molecular Modeling and Chemistry of Natural Compounds, Institute of Molecular Theranostics, Scientific and Technological Park of Biomedicine, I.M. Sechenov First Moscow State Medical University (Sechenov University), 119048 Moscow, Russia; eroshchenko_n_n@staff.sechenov.ru (N.E.); phenolyat@gmail.com (E.D.); 3Laboratory of Pathophysiology Federal Research and Clinical Center of Physical-Chemical Medicine of Federal Medical Biological Agency, 119435 Moscow, Russia; 4Department of Cardiovascular Surgery, Institute for Professional Education, I.M. Sechenov First Moscow State Medical University (Sechenov University), 119048 Moscow, Russia; komarov_r_n@staff.sechenov.ru (R.K.); dzyundzya_a_n@staff.sechenov.ru (A.D.); 5Department of Analytic Chemistry, Faculty of Chemistry, Lomonosov Moscow State University, 119991 Moscow, Russia; 6Department of Cardiology, Functional and Ultrasound Diagnostics, N.V. Sklifosovsky Institute for Clinical Medicine, I.M. Sechenov First Moscow State Medical University (Sechenov University), 119048 Moscow, Russia; agishm@list.ru; 7Moscow State Healthcare Institution, City Clinical Hospital №1, 8 Leninsky Ave., 119049 Moscow, Russia

**Keywords:** atherosclerosis, lipidome, stroke, brachiocephalic arteries, atherosclerotic plaque, triacylglycerides, lipidomics, mass spectrometry

## Abstract

According to the World Health Organization, ischemic stroke is the second leading cause of death in the world. Frequently, it is caused by brachiocephalic artery (BCA) atherosclerosis. Timely detection of atherosclerosis and its unstable course can allow for a timely response to potentially dangerous changes and reduce the risk of vascular complications. Omics technologies allow us to identify new biomarkers that we can use in diagnosing diseases. This research included 90 blood plasma samples. The study group comprised 52 patients with severe atherosclerotic lesions BCA, and the control group comprised 38 patients with no BCA atherosclerosis. Targeted and panoramic lipidomic profiling of their blood plasma was carried out. There was a statistically significant difference (*p* < 0.05) in the values of the indices saturated fatty acids (FAs), unsaturated FAs, monounsaturated FAs, omega-3, and omega-6. Based on the results on the blood plasma lipidome, we formed models that have a fairly good ability to determine atherosclerotic lesions of the brachiocephalic arteries, as well as a model for identifying unstable atherosclerotic plaques. According only to the panoramic lipidome data, divided into groups according to stable and unstable atherosclerotic plaques, a significant difference was taken into account: *p* value < 0.05 and abs (fold change) > 2. Unfortunately, we did not observe significant differences according to the established plasma panoramic lipidome criteria between patients with stable and unstable plaques. Omics technologies allow us to obtain data about any changes in the body. According to our data, statistically significant differences in lipidomic profiling were obtained when comparing groups with or without BCA atherosclerosis.

## 1. Introduction

According to the World Health Organization (WHO), ischemic stroke is the second leading cause (11% among all) of death in the world [1]. As stated by the Russian registry, the annual stroke incidence is 3.28 (4.15; 2.74) per 1000, while mortality is 0.96 (1.18; 0.81) per 1000 in the population. Ischemic stroke is five times more frequent than intracranial bleeding, and in 15–20% of cases, it is caused by brachiocephalic artery (BCA) atherosclerosis, which sometimes additionally increases the risks due to it being asymptomatic [2,3,4]. Ultrasound examination, which is usually administered either by neurologists in cases of neurological symptoms or cardiologists to verify arterial hypertension target organ lesions or to rule out a diagnosis of multifocal atherosclerosis when other arteries are damaged (coronary arteries, peripheral arteries of the lower limbs), is the most widespread diagnostic method for brachiocephalic artery atherosclerosis. Contrast-enhanced computed tomography (CT) of the brachiocephalic arteries is less common. Detecting vulnerable plaques as a risk factor for stroke is one of the most sufficient advantages of computed tomography; however, CT is an expensive, time-consuming procedure which also requires high-cost equipment and qualified medical staff. Moreover, during CT, a patient undergoes certain radiation and contrast drug exposure.

Plasma total fatty acid levels have been linked to the pathogenesis of atherosclerosis and are being studied as potential biomarkers for the disease.

Saturated fatty acids (SFAs) are involved in various biological processes that contribute to the development and progression of this cardiovascular disease. Saturated fatty acids, such as palmitic acid, induce endothelial dysfunction by increasing the expression of adhesion molecules, cytokines, and inflammatory proteins, which contribute to the development of atherosclerosis [5,6]. Also, SFAs affect cholesterol metabolism by increasing low-density lipoprotein (LDL) cholesterol levels, a major risk factor for atherosclerosis. They also influence macrophage cholesterol homeostasis, leading to increased uptake of oxidized LDL and foam cell formation, which are critical steps in plaque development [7].

While saturated fatty acids (SFAs) are often associated with negative health impacts, recent studies suggest that certain types and levels of SFAs in blood plasma can have positive health effects. Elevated levels of arachidic acid (C20:0), behenic acid (C22:0), and lignoceric acid (C24:0) are linked to a reduced risk of incident diabetes. These fatty acids are associated with lower triglyceride and palmitic acid levels, which may mediate their protective effects against diabetes [8,9].

Plasma monounsaturated fatty acids play a complex role in atherosclerosis. While they can reduce inflammation, they also contribute to lipid profiles and cholesterol ester compositions that can promote plaque formation. The Atherosclerosis Risk in Communities (ARIC) study found that higher plasma levels of MUFAs were associated with increased carotid intima–media thickness, an indicator of atherosclerosis. This relationship persisted independently of other risk factors like age, smoking, and LDL cholesterol [10]. A study on LDL receptor-null, human ApoB100-overexpressing transgenic mice showed that diets high in MUFAs (cis and trans) increased aortic atherosclerosis. A MUFA-rich diet significantly elevated LDL cholesterol and very low-density lipoprotein cholesterol levels, contributing to plaque formation [11]. In animal models, high levels of dietary MUFAs led to significant accumulation of cholesteryl oleate in plasma lipoproteins, which is predictive of atherosclerosis. This was particularly evident in hepatic secretion patterns, linking MUFA intake to increased coronary artery atherosclerosis [12].

Plasma HUFAs, particularly omega-3 fatty acids such as EPA and DHA, play a protective role in cardiovascular health by reducing inflammation and oxidative stress. Plasma n-3 and n-6 fatty acids are inversely associated with inflammatory markers such as C-reactive protein (CRP), interleukin-6 (IL-6), and Tumor Necrosis Factor (TNF-alpha). Conversely, a high n-6/n-3 ratio correlates positively with these markers, indicating higher inflammation and coagulation risks [13].

Thus, nowadays, there is no safe and effective screening method which would not only help to detect brachiocephalic artery atherosclerosis but to identify the groups of patients of high risk who have unstable atherosclerotic lesions.

The aim of this research is to study the features of the blood plasma lipidomic profile in unstable brachiocephalic arteries.

## 2. Materials and Methods

The material (blood plasma samples, atherosclerotic plaques) was collected in the Cardiology Clinic of the University Clinical Hospital № 1, I.M. Sechenov First Moscow State Medical University (Sechenov University) from September 2022 to October 2023. The study was conducted in accordance with the Declaration of Helsinki and approved by the local ethics committee of I.M. Sechenov First Moscow State Medical University (Sechenov University), Protocol Code №16-22, 1 September 2022, registered on ClinicalTrials.gov—NCT05680935.

The present study was essentially an exploratory study with a very small number of patients. The main aim was to determine the association of the lipidome with the presence or absence of atherosclerosis and to select the main features, but not to create “prognostic” models.

Associations and assessments of potential confounding represent a rather ambitious goal, which we would like to address with a larger sample size. In the present, very limited study, we recognize that the groups differ objectively by sex and age, but we cannot state how these traits appear to be related to the lipidome. Accordingly, we removed all factors other than the lipidome from the analysis in order to identify an applicable range of substances associated with atherosclerosis or instability.

Once this circle of substances is identified, we can hypothesize about their association with sex, age and design studies to assess the relationships of all the putative confounders.

A total of 90 blood plasma samples were collected. The first group contained data from 52 patients with severe atherosclerotic lesions, while the second group contained data from 38 patients with no BCA atherosclerosis. The age medians of the 1st and 2nd groups were 68.0 [63.0;72.3] and 49.5 [39.5; 58.0], respectively. Men predominated in both groups. Smoking, arterial hypertension, diabetes mellitus, ischemic vascular complications, and chronic kidney disease were seen in the group of patients with BCA atherosclerosis more frequently (Table 1).

### 2.1. Sample Description

The analysis included data from 52 patients with atherosclerosis of the brachiocephalic arteries who underwent elective surgery—carotid endarterectomy (all patients are alive)—and 38 patients without atherosclerosis of the brachiocephalic arteries. In all patients, 20.0 mL of blood was collected from their antecubital vein on an empty stomach. After this, tubes with ethylenediaminetetraacetic acid (EDTA) containing blood were centrifuged (ELMI centrifuge, centrifuge model SM-6M, Riga, Latvia) once at 1000× *g* for 10 min to sediment the cells; then, ¾ of the plasma volume was taken from above, so as not to capture cells, and put into a new empty tube. The plasma was centrifuged again at 2500× *g* for 15 min to sediment the platelets. The supernatant was aliquoted into 1.0 mL Eppendorf tubes. The material was then frozen (Thermo Fisher Scientific freezer TSX50086A Ultra-Low temperature Freezer −86 °C, Waltham, MA, USA) and stored at −80 °C to −70 °C.

Atherosclerotic plaques were obtained during a planned carotid endarterectomy. The protocol for obtaining samples of the atherosclerotic plaques: The atherosclerotic plaque obtained during surgery (carotid endarterectomy) is cut in half, and the adjacent part of the intima is cut off. One half of the plaque is placed in a tube with 10% neutral buffered formalin solution and sent for histological examination. The second half is placed in tubes with RNAprotect Tissue Reagent (QIAGEN, Venlo, Netherlands), cooled at +2 °C +4 °C, and then frozen (Thermo Fisher Scientific Ultra-Low temperature freezer) and stored at −80 °C to −70 °C.

After histological and morphological examination of the plaques, the following indicators were assessed: the lipid core, inflammatory infiltrate, neoangiogenesis, the fibrous cap, calcifications, ulcerations, parietal thrombosis, hemorrhages. This made it possible to distinguish 2 subgroups in the group with atherosclerosis of the brachiocephalic arteries: patients with a stable and unstable course of atherosclerosis. A total of 24 samples were classified into the group with unstable atherosclerotic plaques and 28 samples into the group with stable plaques.

### 2.2. Blood Sample Preparation for Target Lipidomic Analysis

The HPLC-MS/MS data on the composition of the fatty acids in the plasma were processed using GraphPad Prism 8.0.1 software. The values obtained for each sample were considered unpaired and consistent, with confirmation by the ROUT outlier test (ROUT (Q = 1%)). The normality of the distribution for all the sample groups was assessed and confirmed. Thereafter, the data were analyzed using a one-way ANOVA. The statistical significance of the results was determined by a two-sided *p*-value of less than 0.05.

Blood sample preparation for the lipidomic analysis involved the steps described below. The samples are safely stored at a temperature of −80 °C before the analysis and later undergo careful defrosting on ice. Two hundred and fifty μL of plasma is collected from each sample and transferred into Eppendorf tubes. Five μL of internal standard is added to these aliquots. The next step is adding 200 μL of methanol and 750 μL of heptane to the samples, and then they are placed into an orbital shaker for 10 min. As soon as the incubation of the samples ends, they are subjected to centrifugation at the maximum speed for 10 min. Six hundred and fifty μL of supernatant is collected and transferred into Eppendorf tubes. The obtained samples are lyophilized until dry and then reconstructed into 50 μL of B phase. After that, the samples undergo centrifugation again, and 40 μL of supernatant is transferred into marked vials with screw caps and submitted for analysis.

Blood sample preparation for metabolite detection was conducted as follows: a 100 μL aliquot of each blood plasma sample (calibrator or quality control sample) is mixed with 50 μL of isotope-labeled internal standard (ISTD) (D7-Arg, 1.55 μm) and 40 μL of methanol in a microtiter plate for protein precipitation. After 10 min of incubation, the microtiter plate is centrifuged for 5 min at 13,000× *g*. Thereafter, 40 μL of supernatant is transferred into a falcon tube and mixed with 40 μL of water; then, 1 μL of the obtained solution is entered into an LC/MS/MS system.

### 2.3. Reagent for Target Lipidomic Analysis

The fatty acid methyl ester standards were purchased from Larodan (Sweden) and contain the following saturated fatty acids: C12:0, C14:0, C16:0, C17:0 (IS), C18:0, C20:0, C22:0, C24:0; omega-7 unsaturated fatty acids: C16:1 cis; omega-9 unsaturated fatty acids: C18:1 cis, C20:1 cis, C24:1 cis; trans fatty acids: C16:1 trans, C18:1 trans, C18:2 trans; omega-6 unsaturated fatty acids: C18:2 n- 6, C18:3 n- 6, C20:2 n- 6, C20:3 n- 6, C20:4 n- 6, C22:4 n- 6, C22:5 n- 6; and omega- 3 unsaturated fatty acids: EPA C20:5 n-3, DHA C22:6 n-3.

The solvents used were methanol (Merck, LiChrosolv®, hypergrade for LC-MS, Darmstadt, Germany), acetonitrile, and isopropanol (for HPLC gradient, AppliChem GmbH, Darmstadt, Germany). Deionized water for the HPLC was obtained using the Merck Milli-Q Advantage A10 apparatus (Merck KGaA, Darmstadt, Germany).

### 2.4. Methodic HPLC-MS/MS

The procedure for plasma sample preparation and analysis is described in Eroshchenko, N. N. et al., 2023 [14]. In brief, 40 μL of blood plasma was placed in a 2 mL Eppendorf safe-lock tube. Then, 50 μL of an internal standard solution of methyl heptadecanoate (C17:0) at a concentration of 500 μg mL^−1^ in methanol was added. After that, 910 µL of methanol with BHT 200 μg mL^−1^ was added, and the solution was mixed in a shaker for 30 min at 2000 rpm. The sample was centrifuged for 10 min at 19,000× *g*. Then, 500 μL of the supernatant was transferred into a 1.5 mL glass crimp vial, and 500 μL of a 15% solution of boron trifluoride (BF3) in methanol was added. The vial was crimp-sealed, mixed, and placed in a thermostat at 100 °C for 90 min. After incubation, the samples were transferred into a refrigerator at +4 °C for 30 min. The supernatant was transferred into a 1.5 mL glass vial with a screw cap. Deionized water (500 µL) was added, and the resulting samples were analyzed using the HPLC-MS/MS method.

The Liquid Chromatograph LC-20AD Prominence (Shimazdu, Japan) was used for chromatographic separation. Eluent A was prepared by adding 0.1% aqueous formic acid to deionized water and a 5 mM ammonium formate solution. Eluent B was a mixture of acetonitrile and isopropanol (1:1) with the addition of 1% eluent A. Detection was performed on a Sciex 4500 QTRAP mass spectrometer (Sciex, Toronto, Canada) using electrospray ionization in positive mode. The results of the sample analysis were processed using Analyst 1.6.3 and MultiQuant 3.0 software (Sciex, Toronro, Canada). Ionization source parameters: Curtain Gas (CUR)—35; Collision Gas (CAD)—Medium; Ion Spray Voltage (IS)—5500; Temperature (TEM)—0; Ion Source Gas 1 (GS1)—50; Ion Source Gas 2 (GS2)—50. The standard protocol of panoramic lipid plasma analysis was used

The data were obtained using multiple reaction monitoring (MRM) in positive mode with [M + NH4]+ ions. List of MRM (Q1/Q3): C12:0 (232.3/215.3), C14:0 (260.3/243.3), C16:0 (288.3/271.3), C18:0 (316.2/299.2), C20:0 (344.3/327.3), C22:0 (372.3/355.3), C24:0 (400.3/383.3), C17:0 IS (302.3/285.3), C16:1 cis (286.3/269.3), C18:1 cis (314.3/297.3), C20:1 cis (342.3/325.3), C24:1 cis (398.3/381.3), C16:1 trans (286.3/269.3), C18:1 trans (314.3/297.3), C18:2 trans (312.3/295.3), C18:2 n-6 (312.3/295.3), C18:3 n-6 (310.3/293.3), C20:2 n-6 (340.3/323.3), C20:3 n-6 (338.3/321.3), C20:4 n-6 (336.3/319.3), C22:4 n-6 (364.3/347.3), C22:5 n-6 (362.3/345.3), EPA C20:5 n-3 (334.3/317.3), DHA C22:6 n-3 (360.3/343.3).

### 2.5. Sample Preparation

Prior to the start of the sample preparation, the samples were stored at −80 °C. The samples were thawed on ice. After defrosting, the samples were placed on a vortex for 10 s. Aliquots of 100 µL were taken from the samples and transferred into separate Eppendorf tubes. Next, 400 µL of 1-butanol containing an internal standard was added to the samples. The samples were placed in a shaker for 10 min at 1200 rpm at room temperature. The samples were then centrifuged at 13000 rpm for 10 min at 5 C. After the end of centrifugation, 200 microliters of the organic layer were taken from the samples and transferred for analysis.

Twenty-five percent of the samples studied were prepared in two bio-repeats.

### 2.6. Analysis Conditions

A Sciex 6600QTOF time-of-flight mass spectrometer (Sciex, Toronto, Canada) with a calibrant delivery system (CDS) with an Agilent 1290 Infinity II liquid chromatograph was used for the analysis. Ion source settings: TEM = 300 °C; GS1 = 55; 2 = 55; CUR = 30; IS = 5500. Ion detection was carried out in positive ionization mode for the samples in the TOFMS in the range of 350–1700 *m*/*z*.

Chromatographic separation of the test sample components was carried out in RPLC chromatography mode using the Waters ACQUITY C8 chromatographic column (2.1 × 100 mm 1.7 μm): phase A (water:acetonitrile (4:6); 10 mM ammonium formate); phase B (acetonitrile:isopropanol (1:9); 10 mM ammonium formate); sample input volume: 2 µL. Chromatographic gradient: 0 min 10% B; 0.15 min 10% B; 2 min 30% B; 2.5 min 48% B; 11 min 65% B; 12 min 99% B; 14 min 99% B; 14.1 min 10%B; 16 min 10% B; the flow rate was 0.25 mL/min, and the thermostat temperature was 55 °C.

The samples in the analytical series were randomized and examined in two technical repeats per sample.

The initial solution for the quality control samples was obtained by combining aliquots from a common sample pool. Next, the initial solution for the quality control samples was divided into aliquots and frozen. Before starting the analysis, the required number of quality control samples was thawed and analyzed as part of the analytical series; at least 10% of the analytical series accounted for the quality control samples.

### 2.7. Processing the Results

To process the results, the SCIEX PeakView 2.2 + SCIEX MasterView 1.1 software was used with Skyline 23.1.0 and MSDIAL 5.3.240311. MSDIAL software (an open source software for untargeted metabolomics and lipidomics data, https://www.lipidmaps.org) with a generated lipid library and the MS-DIAL LipidBlast library (ver. 68) were used for lipid annotation.

### 2.8. Statistics

Statistical analysis was carried out using the programming languages R v.4.2 and Python v.3.10 [^R].

Normality (using the Shapiro–Wilk test), means, standard deviation, medians, interquartile ranges, 95% confidence intervals, and minimum and maximum values were calculated for quantitative indicators. The proportion and absolute number of the values were calculated for categorial and qualitative characteristics.

Comparative analysis for normally distributed quantitative characteristics was carried out based on Welch’s *t*-test (2 groups), while for non-normally distributed quantitative characteristics, the Mann–Whitney U test was used (2 groups).

Comparative analysis for categorial and qualitative characteristics was carried out using Pearson’s square method; if inapplicable, Fisher’s exact test was used.

The mathematical modeling pipeline was used to evaluate the lipidome affecting on endpoints such as either being in the study group or the control group and atherosclerotic plaque stability or instability. Exploratory data analysis included Ward clustering and K-means clustering using data on the lipidome. The data could be divided into 2 clusters among both groups, with subsequent correlation of the clusters with the presence/absence of atherosclerotic lesions. Afterwards, clustering into 2 groups was performed in the group with atherosclerosis to try to relate it to the stability or instability of the atherosclerotic plaques. 

## 3. Results

### 3.1. Targeted Lipidomic Analysis

An understanding of the variation in FA composition can provide insights into the biochemical differences associated with plaque stability, which is crucial for developing targeted therapeutic strategies. The provided graph (Figure 1) depicts the percentage composition of distinct types of fatty acids (FAs) in relation to the total FA content. The y-axis, which is labeled with percentage values, quantifies the relative abundance of each fatty acid category or individual fatty acid.

Firstly, we examine the levels of individual fatty acids. In general, individuals with control levels of various fatty acids exhibit higher levels than those with plaques (both unstable and stable). C18:1 cis and C16:0 have long been associated with the development of inflammatory diseases, including cardiovascular disease, based on their presence in the blood. Elevated plasma levels of C16:0 and C18:1 cis are associated with an increased risk and progression of atherosclerosis. These fatty acids contribute to the formation of plaques through their roles in lipid metabolism, inflammation, and oxidative stress [6]. Conversely, there is also evidence that elevated plasma saturated acid levels were associated with a lower risk of coronary heart disease in population-based studies [15].

Our data are consistent with the aforementioned elevated values for C18:1 fatty acid, although the elevated content of palmitic acid in the control samples is rather unusual. Our data are consistent with previous studies that have demonstrated plasma levels of 18:1 cis, along with other fatty acids, are potentially related to oxidative stress and lipid peroxidation in coronary atherosclerosis. This highlights the complex roles of different fatty acids in the disease process [16].

Another marker acid was C20:4 n-6 (arachidonic acid), an omega-6 polyunsaturated fatty acid (PUFA). Arachidonic acid was higher in the healthy individuals compared to those with atherosclerosis. This is particularly interesting since arachidonic acid is often associated with inflammatory processes [17]. However, it appears that in this context, higher arachidonic acid levels might be associated with protective or regulatory mechanisms that counteract atherosclerosis development when balanced appropriately. Although the difference in the C20:4 n-6 levels in the plasma was not as significant as in other cases, it is noteworthy that the difference in the patient samples with stable plaques was more consistent.

The analysis revealed that healthy individuals have a higher proportion of saturated fatty acids compared to patients with atherosclerosis. This finding suggests that higher levels of saturated fatty acids may not necessarily correlate with increased atherosclerosis risk [6]. Healthy individuals were found to have lower levels of unsaturated fatty acids. This includes lower levels of oleic acid (C18:1 cis), a monounsaturated fatty acid (MUFA). Lower oleic acid levels in healthy individuals might indicate that not all unsaturated fats are equally protective against atherosclerosis, and the specific types and balances of fatty acids play crucial roles. Our data are consistent with previous studies, where it was found that in hyperlipidemic patients, a decrease in saturated fatty acids and an increase in unsaturated fatty acids are observed [18].

The analysis shows no statistical difference in the PUFA levels between healthy individuals and those with atherosclerosis. This indicates that the total PUFA content alone may not be a decisive factor in the development or prevention of atherosclerosis. Healthy individuals have slightly higher levels of HUFAs. This suggests that higher HUFA levels might contribute to better cardiovascular health, potentially through anti-inflammatory or other protective mechanisms [19].

The levels of omega-6 and omega-3 fatty acids were not statistically different between the two groups. This suggests that the balance of these essential fatty acids alone does not explain the differences in atherosclerosis development, highlighting the importance of other fatty acid types and their interactions in disease progression.

Due to the unknown role of each quantitative indicator and all of them together in relation to such endpoints as group membership and the stability/instability of atherosclerotic plaques in the main group, cluster analysis was performed only on the lipidome data.

The quality of the clustering was assessed using ROC analysis, where membership in the main or control group was used as a reference.

We tried different clustering algorithms and assigned BCA, stable/unstable atherosclerotic plaques, and groups with/without atherosclerosis.

### 3.2. Cluster Analysis

The best separation into clusters corresponding to the outcomes was achieved using Ward’s algorithm. The clustering corresponded to belonging to a group.

In terms of the results of dividing all of the indicators of the targeted and panoramic lipidome into two clusters using Ward’s algorithm, the division into clusters correlated well with group membership: AUC 0.82 [0.65;0.94], Sens 0.80 [0.50;1.00], Spec 0.85 [0.75;0.93], NPV 0.50 [0.23;0.75], PPV 0.96 [0.89; 1.00].

The characteristics of both classes, clinical, anamnestic, and others, are presented in the Appendix A.

One of the clusters is, in fact, completely represented by patients without atherosclerosis. Separation is provided by both types of lipidome (targeted and panoramic). We conducted a similar analysis in the group with BCA atherosclerosis, and when divided into two clusters, a weak relationship between clustering and plaque stability/instability was noticed. This can be interpreted as the inability to use these signs to characterize the atherosclerotic process as stable or unstable: AUC 0.55 [0.46; 0.63], Sens 0.08 [0.0; 0.21], Spec 0.82 [0.68; 0.94], NPV 0.29 [0.0; 0.63], PPV 0.51 [0.36; 0.65].

Due to the small number included in the study and the significant number of potential predictors, non-standard approaches were used to select the predictors most strongly associated with the outcome (group membership and stability/instability of atherosclerotic plaques). For each LASSO regression model created in the leave-one-out cross validation process, we selected the 10 predictors from both lipidomic analysis of all patients with the highest absolute coefficients. We obtained 10 predictors related to the panoramic lipidome (Table 2). Then, we averaged the coefficients for each model and formed a list of predictors with the highest averaged coefficients.

In the only panoramic lipidomic analysis of all the patients, the following 10 predictors were identified (Table 3).

For the stable/unstable groups (52 patients, of which 24 showed signs of plaque instability) from both lipidomic analyses, we have the next ten predictors with the highest averaged absolute coefficients (Table 4). 

### 3.3. Panoramic Lipidomic Analysis

When analyzing the overall lipidome, 423 lipids were identified in patients with or without BCA atherosclerosis (Figure 2).

The batch effect, which was possibly associated with the samples’ storage duration, was noticed during the overall lipidome analysis. Thus, ten blood plasma samples with the same storage time from the second group were included in the analysis. When conducting target lipidome analysis, no batch effect was noticed, which is why all the samples were used for the research. Most likely, this effect was caused by ceramide, dimetlarginine, and other lipid oxidation over the course of time, so comparison of samples with different storage durations cannot provide objectivity. It is worth mentioning that fatty acids are stable enough and are not subject to rapid oxidation when stored at −80 degrees.

In the groups divided by the stability and instability of the atherosclerotic plaques, a significant difference was also considered: *p*-value < 0.05 and abs(fold change) > 2. However, even in this case, we did not observe any significant differences in terms of the established criteria for the blood plasma lipidome between the patients with stable and unstable plaques.

## 4. Discussion

Nowadays, clinical application of the standard lipidome is usedin cardiological practice because of the proven decrease in adverse outcomes due to therapy aiming to reduce the levels of atherogenic lipids. At the moment, we do not use all the capabilities of the lipidome in stratifying the risk of cardiovascular complications. Panoramic and targeted lipidomes may help to more clearly stratify these patients by risk of complications. Lipidome profiling may also be important in various diseases. Nevertheless, recent decades have shown the remarkably increased interest of the scientific community in studying the lipidome within a wide range of pathologies and for various fluids of the body [20]. Australian scientists have already identified some changes in the lipidome which help to differentiate Alzheimer’s disease with a high level of accuracy [21]. Other research has shown the negative effect of lipid metabolism changes in diabetes mellitus patients on the bloodstream [22]. Fifteen lipids were associated with a high risk of type 2 diabetes mellitus developing in patients with coronary heart disease in 5 years in the CARDIOPREV research; moreover, a new classifier showing high efficacy in glucose metabolism disorder prognosis was created based on this information [23]. Furthermore, 21 lipids associated with diabetes mellitus were also found in research with a 16-year-long observation period by Miao G. Several scientific works have demonstrated lipidomic features specific to chronic kidney disease, an elevated risk of atrial fibrillation, and systemic lupus erythematosus [24].

This study is dedicated to studying the lipidome characteristics in brachycephalic atherosclerosis, including cases of unstable plaques. There is a great variety of works aimed at studying the characteristics of lipid metabolism in diverse manifestations of atherosclerosis. According to our data, in patients with significant BCA atherosclerosis, when comparing groups with stable or unstable atherosclerotic plaques, no significant differences in the lipidomic profiling of their blood plasma were detected. Most likely, this is due to the presence in these same patients of other atherosclerotic plaques that contribute to the lipidome.

In a study by Sojo, L. et al., the association of subclinical atherosclerosis in type 1 diabetes mellitus patients with a spectrum of lipids was studied; the most significant positive associations for plaques localized in the brachiocephalic arteries were obtained for sphingomyelin [25]. Significant association between plasma lipids and the absence of carotid plaques in patients from the Chinese population was found by Liu Y., and it was discovered for HDL-C, non-HDL-C, TC/HDL-C, LDL-C/HDL-C; HDL-C, LDL-C, non-HDL-C, TC/HDL-C, and LDL-C/HDL-C that their levels correlated with carotid plaque absence [26].

In research by You, Q. et al., the characteristics of the lipidome were studied among people with atherosclerotic lesions of the main (20 people) and small (20 people) brachycephalic vessels in comparison with healthy volunteers (14 people). An increase in the ceramides Cer (d36:3), Cer (d34:2), Cer (d38:6), Cer (d36:4), and Cer (d16:0/18:1) characterized damage to the main vessels; sphingomyelin SM (d34:1) and ceramide Cer (d34:2), Cer (d36:4), Cer (d16:0/18:1); Cer (d38:6), Cer (d36:3), and Cer (d32:0) level increases were typical for internal vessel lesions; and when comparing the two groups presented above, a rise in Cer (d36:4) and SM (d34:1) was detected [27].

A study be Nieddu, G. et al. showed that lipids belonging to the group of phosphatidylethanolamine (PE), sphingomyelin (SM), and diacylglycerol (DG) can be used to differentiate plaque types: the analysis revealed significant changes for LDL PE (38:6), SM (32:1), and SM (32:2) when comparing vulnerable and stable atherosclerotic lesions [28].

Slijkhuis, N. B. et al. compared the blood plasma lipidome and carotid plaque lipidome after endarterectomy. Both lipidomes were rich in ceramides (37% for plasma and 63% for plaques); cholesteryl oleate and cholesteryl linoleate were most common, and cholesteryl oleate was mostly present in plaques. The free fatty acids FFA (16:0) and FFA (18:1) were detected at similar frequencies in both plasma and plaques lipidome, although FFA (16:0) was more common in the plaques lipidome. FFA (18:1), FFA (18:2), FFA (20:4), and FFA (22:4) were identified in areas of plaques rich in macrophages, which, according to the authors, indirectly indicates their pro-inflammatory role. Phosphatidylcholines were more common in plasma, but two of them, PC (32:0) and PC (34:1), predominated in plaques [29].

## 5. Conclusions

The blood lipidome parameters revealed in our study can be used in the detection of atherosclerotic lesions of the brachiocephalic arteries. Triacylglycerides had the highest representation (14:0, 18:0, 18:3, 54:7, 55:1), followed by diacylglycerides (13:0, 41:3) and phosphatidylcholine (18:1, 18:2, 36:3, 37.3, 40:8). However, at the moment, there are no indicators of the targeted or panoramic plasma lipidome with a sufficient level of evidence that would help characterize the unstable course of atherosclerosis.

This analysis reveals that healthy individuals tend to have higher levels of certain saturated fatty acids (e.g., palmitic acid) and arachidonic acid while having lower levels of MUFAs (e.g., oleic acid). These findings highlight the complexity of fatty acid interactions in cardiovascular health and suggest that a simple categorization of fatty acids as “good” or “bad” may be insufficient. Instead, the balance and specific types of fatty acids are crucial in understanding and managing the risk of atherosclerosis.

This analysis found no significant differences in the plasma fatty acid profiles between patients with stable and unstable plaques. This indicates that the fatty acid composition may not be a distinguishing factor for plaque stability and that other factors, such as inflammatory markers or mechanical stress, might play more significant roles in plaque stability. Additionally, the lack of differences in the omega-6 and omega-3 levels, as well as within the atherosclerosis patient group, underscores the need for a nuanced understanding of lipid metabolism and its impact on cardiovascular disease.

The resulting models allow us to identify a number of factors associated with atherosclerotic changes. The constructed models were of moderate quality (given the extremely small number of patients in the study and its hypothesis-generating nature), which shows the need for further research to assess the diagnostic properties of the lipidome in relation to the presence and progression of atherosclerosis.

The study of lipidomics makes a great contribution to understanding the development of this disease, expands the possibilities of diagnosis, and allows us to use the principles of personalized medicine but requires further research.

## Figures and Tables

**Figure 1 biomedicines-12-01279-f001:**
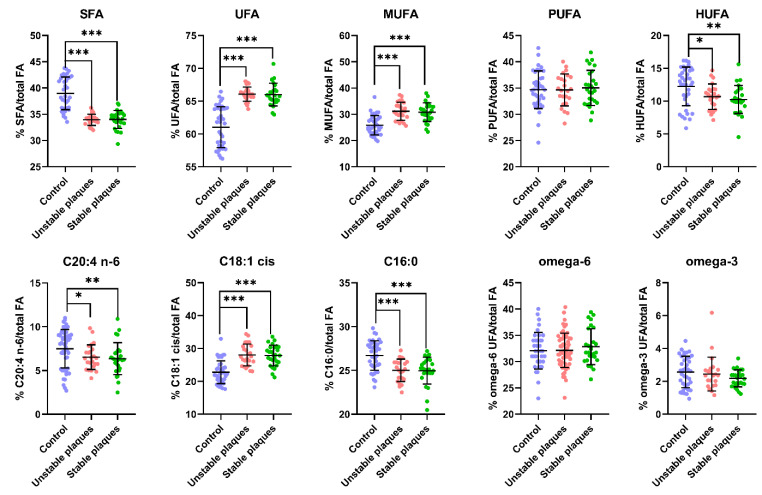
The impact of the formation of plaques on the levels of fatty acids in the blood plasma. The data presented here represent the means of normalized percentages of a parameter ± S.D. Statistically significant differences are indicated by * (*p* < 0.025), ** (*p* < 0.005), and *** (*p* < 0.0005), which were evaluated by ANOVA.

**Figure 2 biomedicines-12-01279-f002:**
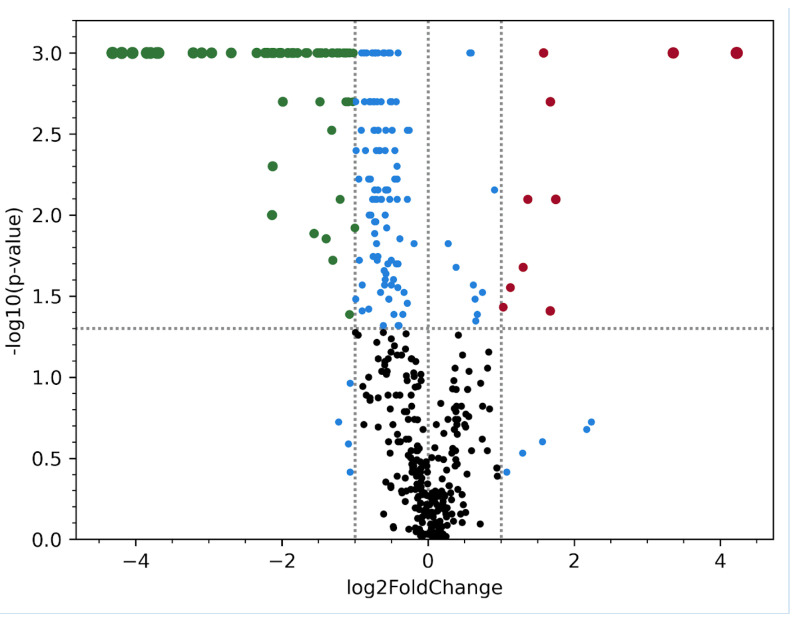
Volcano plot of lipidomics illustrating compounds differing between the patients with BCA and healthy control groups. Each dot represents 1 of 453 compounds. The *x*-axis represents log2 (Fold change), and the *y*-axis represents −1og10 (*p*-value). The two lines parallel to the *y*-axis are x = −1 and x = 1. The points to the left of x = −1 and to the right of x = 1 are compounds with differences > two-fold. The line parallel to the *x*-axis is y = 1.30. Red points represent 10 lipid compounds upregulated significantly in BCA group, based on fold-change > 2 and *p* < 0.05 (Mann–Whitney U test). Green points represent 72 lipid compounds downregulated significantly in BCA group based on same criteria. Blue dots represent metabolites for which the change in abundance between experimental groups is not significant according to one of the significance criteria. Black dots represent lipids with no significant differences.

**Table 1 biomedicines-12-01279-t001:** Characteristics of groups.

	Control Group	Study Group	
sex			0.030
female	17 (44.7%)	12 (23.1%)	
male	21 (55.3%)	40 (76.9%)	
age, years			<0.001
Mean ± Standard Deviation	49.0 ± 11.5	67.2 ± 8.3	
Median and [25%; 75%]	49.5 [39.5; 58.0]	68.0 [63.0; 72.3]	
weight, kg			0.839
Mean ± Standard Deviation	80.1 ± 13.3	80.7 ± 11.7	
Median and [25%; 75%]	79.0 [72.0; 89.8]	79.0 [70.8; 90.0]	
height, m			0.930
Mean ± Standard Deviation	1.73 ± 0.1	1.68 ± 0.1	
Median and [25%; 75%]	1.7 [1.7; 1.8]	1.7 [1.7; 1.8]	
BMI			0.659
Mean ± Standard Deviation	27.0 ± 4.0	27.4 ± 5.1	
Median and [25%; 75%]	26.9 [25.1; 29.3]	25.9 [23.9; 30.1]	
smoking			0.001
0	32 (84.2%)	27 (51.9%)	
1	6 (15.8%)	25 (48.1%)	
diabetes mellitus			0.021
0	36 (94.7%)	40 (76.9%)	
1	2 (5.3%)	12 (23.1%)	
arterial hypertension			<0.001
0	24 (63.2%)	5 (9.6%)	
1	14 (36.8%)	47 (90.4%)	
coronary artery disease			<0.001
0	36 (94.7%)	34 (65.4%)	
1	2 (5.3%)	18 (34.6%)	
ischemic stroke			<0.001
0	38 (100.0%)	36 (69.2%)	
1	0 (0.0%)	16 (30.8%)	
heart failure			0.135
0	38 (100.0%)	48 (92.3%)	
1	0 (0.0%)	4 (7.7%)	
chronic kidney disease, stage			<0.001
0	32 (84.2%)	6 (11.5%)	
1	3 (7.9%)	2 (3.8%)	
2	3 (7.9%)	21 (40.4%)	
3a	0 (0.0%)	21 (40.4%)	
3б	0 (0.0%)	2 (3.8%)	

**Table 2 biomedicines-12-01279-t002:** Ten predictors from panoramic and targeted lipidomic analysis with the highest averaged absolute coefficients.

Predictors	AUC	Sensitivity	Specificity	NPV	PPV
TG_54.7_O_TG_18.3_18.3_18.1_O_M..NH4_7.62 PC_40.8_PC_20.4_20.4_M..H_4.86 TG_42.0_TG_14.0_14.0_14.0_M..Na_10.0 TG_42.1_TG_12.0_12.0_18.1_M..Na_9.39 TG_54.1_TG_18.0_18.0_18.1_OA1_M..NH4_12.81 DG_13.0_M..Na_2.95 TG_48.2_TG_16.0_16.1_16.1_M..Na_10.94 TG_48.3_TG_16.1_16.1_16.1_M..Na_10.29 TG_O_55.1_TG_O_19.1_18.0_18.0_M..NH4_12.81 PC_O_36.3_PC_O_18.1_18.2_M..H_5.7	0.86 [0.71; 097]	0.92 [0.85; 0.98]	0.8 [0.5;1]	0.67 [0.36; 0.92]	0.96 [0.9; 1]

**Table 3 biomedicines-12-01279-t003:** Ten predictors from panoramic lipidomic analysis with the highest averaged absolute coefficients.

Predictors	AUC	Sensitivity	Specificity	NPV	PPV
TG_52.4_O_TG_18.2_18.2_16.0_O_M..NH4_8.61 LPC_O_18.1_M..H_2.44, PI_34.2_M..NH4_4.55 TG_54.6_TG_16.0_18.2_20.4_M..NH4_10.85 DG_28.3_OA1_M..Na_0.96 DG_41.3_M..Na_10.88 SM_36.2_O3_M..H_5.93 PI_36.4_M..NH4_4.53 TG_54.7_O_TG_18.3_18.3_18.1_O_M..NH4_7.62 SM_41.3_O2_M..H_6.06	0.98 [0.94; 1]	0.96 [0.87; 1]	1.0 [1; 1]	0.97 [0.89; 1]	1 [1; 1]

**Table 4 biomedicines-12-01279-t004:** Ten predictors from panoramic and targeted lipidomic analyses in stable/unstable plaque groups with the highest averaged absolute coefficients.

Predictors	AUC	Sensitivity	Specificity	NPV	PPV
TG_54.7_O_TG_18.3_18.3_18.1_O_M..NH4_7.62 NAE_22.6_M..H_2.88 PC_O_38.3_PC_O_18.0_20.3_M..H_6.35 C18.2_trans_1_percmol DG_41.3_M..Na_10.88 LPC_22.5_M..H_1.91 LPC_O_18.1_M..H_2.44 PI_34.2_M..NH4_4.55 LPC_18.3_0.0_M..H_1.51 PC_37.3_M..H_5.63	0.58 [0.44; 0.715]	0.625 [0.421; 0.81]	0.536 [0.345; 0.724]	0.625 [0.417; 0.812]	0.536 [0.345; 0.72 ]

## Data Availability

The data can be provided at the official request of the Principal Investigator due to the fact that our local ethics committee does not allow them to be provided openly.

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
