# Peer review of "The Blood Plasma Lipidomic Profile in Atherosclerosis of the Brachiocephalic Arteries"

_biomedicines, 2024, doi:10.3390/biomedicines12061279_

Round 1

Reviewer 1 Report (Previous Reviewer 1)

Comments and Suggestions for Authors

A statement like "However, even in this case, we did not observe any significant differences in terms of the established criteria for the blood plasma lipidome between patients with stable and unstable plaques." (see revised manuscript, line 402) should be added to the abstract. 

Comments on the Quality of English Language

English improved.

Author Response

Thank you for your attention to the manuscript, suggested clarifications and comments. This allowed us to significantly improve the content and presentation of the article.

Reviewer 2 Report (Previous Reviewer 2)

Comments and Suggestions for Authors

This version is much improved. Most of my previous concerns were addressed.

I have no further comments.

Comments on the Quality of English Language

No further comments. Most of my previous questions were addressed.

Thank you!

Author Response

Summary

Thank you very much for taking the time to review this version of manuscript. Thank you for your attention to the manuscript, suggested clarifications and comments. This allowed us to significantly improve the content and presentation of the article.

Best regards,

Anastasiia Lomonosova

This manuscript is a resubmission of an earlier submission. The following is a list of the peer review reports and author responses from that submission.

Round 1

Reviewer 1 Report

Comments and Suggestions for Authors

This manuscript describes the lipidomic profiling of blood plasma samples from patients with or without BCA atherosclerosis. In atherosclerosis patients a significant increase of certain fatty acids could be detected. The aim of investigating the properties of the lipid profile of the blood plasma in order to detect unstable brachiocephalic plaques was not achieved. 

Major

1, The authors used plasma from patients with or without atherosclerosis. In the manuscript analysis of atherosclerosis is only shortly and not clearly described in “Sample description” (page 4, line 83-88). How are the 58 plaques available? How were they detected? Are the patients still alive?

The authors write: “After histological and morphological examinations the plaques were divided into 2 groups: ‘stable’ and ‘unstable’ which means presenting de-stabilization features such as hemorrhage, large lipid core, etc.” It is not clearly described how this is done in living patients.

 2, Ethics for human samples should be added.

 3, Atherosclerosis can be detected using various methods. Lipid changes shown in Fig.1 should be tested now in a second cohort of different patients. However, the detection of unstable, dangerous plaques in plasma would provide additional information. Unfortunately, no differences in the lipidome analysis could be detected here.

Comments on the Quality of English Language

English needs to be greatly improved.

Author Response

Thank you very much for taking the time to review this manuscript. Please find the detailed responses below and the corresponding revisions, corrections highlighted, in track changes in the re-submitted files.

Reviewer 2 Report

Comments and Suggestions for Authors

Dr. Anastasiia Lomonosova et al. assessed blood plasma lipidomic profile in patients with atherosclerotic brachiocephalic arteries (BCA). The authors compared outcome in two groups: Case group of 52 patients with severe atherosclerotic lesions versus control group of 38 patients with no BCA atherosclerosis. Targeted and panoramic lipidomic profiling were carried out using HPLC-MS/MS. Authors found a statistically significant difference between the two groups for the indices of saturated fatty acids (FA), unsaturated FA, monounsaturated FA and omega-3 index, etc.

This is an interesting study, however the authors need to address the following concerns:

1/ It is not clear how the authors stratified the groups by stability and instability of atherosclerotic plaques. How many subjects in each groups and what was the outcome in each group (stable vs. instable plaques)?

2/ Table 1 showed some differences between control and case groups (age, etc.). These potential confounding parameters may have influenced the study outcome and should be taken in consideration especially when establishing predictive models.

3/ Table 1: Interestingly, control and case group were having exactly the same mean/median height (1.7+/- 1)/[1.7-1.8].

4/ For consistency, please use same groups nomination for table 2 as in table 1(control group vs. Case group).

5/ It is highly recommended to provide a clear methodology for clustering selection and steps for creation of predictive models for BCA. This should be included in Methods section.

6/ Data for all selected predictors outcomes (based on group membership, athero. stable/unstable) should be presented in table form (not as text) for better visualization and interpretation of results. Also, data should be presented separately for targeted vs. panoramic lipidomic analysis.

7/ Fig. 1: It not clear what are all the colored dots? What groups of lipids are all the presented data? How to differentiate them? Need legend for the fig.1

8/ The authors stated that the created models have a good diagnostic ability. Authors should be cautious due to small size population and lack of validation in larger cohort. Also, the authors should discuss the limitation of the study including the methodology tools used to generate such predictive models for BCA.

Comments on the Quality of English Language

English proof editing is recommended.

Author Response

(The authors gave the same response as above.)
